# CLARINET: PARALLEL WAVE GENERATION IN END-TO-END TEXT-TO-SPEECH

**Wei Ping**[*]    **Kainan Peng**[*]    **Jitong Chen**[*]

{pingwei01, pengkainan, chenjitong01}@baidu.com

Baidu Research
1195 Bordeaux Dr, Sunnyvale, CA 94089

## ABSTRACT

In this work, we propose a new solution for parallel wave generation by WaveNet. In contrast to parallel WaveNet (van den Oord et al., 2018), we distill a Gaussian *inverse autoregressive flow* from the autoregressive WaveNet by minimizing a regularized *KL divergence* between their highly-peaked output distributions. Our method computes the KL divergence in closed-form, which simplifies the training algorithm and provides very efficient distillation. In addition, we introduce the first text-to-wave neural architecture for speech synthesis, which is fully convolutional and enables fast end-to-end training from scratch. It significantly outperforms the previous pipeline that connects a text-to-spectrogram model to a separately trained WaveNet (Ping et al., 2018). We also successfully distill a parallel waveform synthesizer conditioned on the hidden representation in this end-to-end model. [1]

## 1 INTRODUCTION

Speech synthesis, also called text-to-speech (TTS), is traditionally done with complex multi-stage hand-engineered pipelines (Taylor, 2009). Recent successes of deep learning methods for TTS lead to high-fidelity speech synthesis (van den Oord et al., 2016a), much simpler "end-to-end" pipelines (Sotelo et al., 2017; Wang et al., 2017; Ping et al., 2018), and a single TTS model that reproduces thousands of different voices (Ping et al., 2018).

WaveNet (van den Oord et al., 2016a) is an autoregressive generative model for waveform synthesis. It operates at a very high temporal resolution of raw audios (e.g., 24,000 samples per second). Its convolutional structure enables parallel processing at training by teacher-forcing the complete sequence of audio samples. However, the autoregressive nature of WaveNet makes it prohibitively slow at inference, because each sample must be drawn from the output distribution before it can be passed in as input at the next time-step. In order to generate high-fidelity speech in real time, one has to develop highly engineered inference kernels (e.g., Arık et al., 2017a).

Most recently, van den Oord et al. (2018) proposed a teacher-student framework to distill a parallel feed-forward network from an autoregressive teacher WaveNet. The non-autoregressive student model can generate high-fidelity speech at 20 times faster than real-time. To backpropagate through random samples during distillation, parallel WaveNet employs the mixture of logistics (MoL) distribution (Salimans et al., 2017) as the output distribution for teacher WaveNet, and a logistic distribution based *inverse autoregressive flow* (IAF) (Kingma et al., 2016) as the student model. It minimizes a set of losses including the *KL divergence* between the output distributions of the student and teacher networks. However, one has to apply Monte Carlo method to approximate the intractable KL divergence between the logistic and MoL distributions, which may introduce large variances in gradients for highly peaked distributions, and lead to an unstable training in practice.

In this work, we propose a novel parallel wave generation method based on the Gaussian IAF. Specifically, we make the following contributions:

---

[*]These authors contributed equally to this work. Our method is named after the musical instrument clarinet, whose sound resembles human voice.

[1]Audio samples are in https://clarinet-demo.github.io/

1. We demonstrate that a single variance-bounded Gaussian is sufficient for modeling the raw waveform in WaveNet without degradation of audio quality. In contrast to the quantized surrogate loss (Salimans et al., 2017) in parallel WaveNet, our Gaussian autoregressive WaveNet is simply trained with maximum likelihood estimation (MLE).

2. We distill a Gaussian IAF from the autoregressive WaveNet by minimizing a regularized KL divergence between their peaked output distributions. Our method provides closed-form estimation of KL divergence, which largely simplifies the distillation algorithm and stabilizes the training process.

3. In previous studies, "end-to-end" speech synthesis actually refers to the text-to-spectrogram models with a separate waveform synthesizer (i.e., vocoder) (Sotelo et al., 2017; Wang et al., 2017). We introduce the first text-to-wave neural architecture for TTS, which is fully convolutional and enables fast end-to-end training from scratch. In our architecture, the WaveNet module is conditioned on the hidden states instead of mel-spectrograms (Ping et al., 2018; Shen et al., 2018), which is crucial to the success of training from scratch. Our text-to-wave model significantly outperforms the separately trained pipeline (Ping et al., 2018) in naturalness.

4. We also successfully distill a parallel neural vocoder conditioned on the learned hidden representation within the end-to-end architecture. The text-to-wave model with the parallel vocoder obtains competitive results as the model with an autoregressive vocoder.

We organize the rest of this paper as follows. Section 2 discusses related work. We propose the parallel wave generation method in Section 3, and present the text-to-wave architecture in Section 4. We report experimental results in Section 5 and conclude the paper in Section 6.

## 2 RELATED WORK

Neural speech synthesis has obtained the state-of-the-art results and gained a lot of attention recently. Several neural TTS systems were proposed, including Deep Voice 1 (Arık et al., 2017a), Deep Voice 2 (Arık et al., 2017b), Deep Voice 3 (Ping et al., 2018), Tacotron (Wang et al., 2017), Tacotron 2 (Shen et al., 2018), Char2Wav (Sotelo et al., 2017), and VoiceLoop (Taigman et al., 2018). Deep Voice 1 & 2 retain the traditional TTS pipeline, which has separate grapheme-to-phoneme, phoneme duration, fundamental frequency, and waveform synthesis models. In contrast, Deep Voice 3, Tacotron, and Char2Wav employ the attention based sequence-to-sequence models (Bahdanau et al., 2015), yielding more compact architectures. In the literature, these models are usually referred to as "end-to-end" speech synthesis. However, they actually depend on a traditional vocoder (Morise et al., 2016), the Griffin-Lim algorithm (Griffin and Lim, 1984), or a separately trained neural vocoder (Ping et al., 2018; Shen et al., 2018) to convert the predicted spectrogram to raw audio. In this work, we propose the first text-to-wave neural architecture for TTS based on Deep Voice 3 (Ping et al., 2018).

The neural network based vocoders, such as WaveNet (van den Oord et al., 2016a) and SampleRNN (Mehri et al., 2017), play a very important role in recent advances of speech synthesis. In a TTS system, WaveNet can be conditioned on linguistic features, fundamental frequency ($F_0$), phoneme durations (van den Oord et al., 2016a; Arık et al., 2017a), or the predicted mel-spectrograms from a text-to-spectrogram model (Ping et al., 2018). We test our parallel waveform synthesis method by conditioning it on mel-spectrograms and hidden representation within the end-to-end model.

Normalizing flows (Rezende and Mohamed, 2015; Dinh et al., 2014) are a family of stochastic generative models, in which a simple initial distribution is transformed into a more complex one by applying a series of invertible transformations. Normalizing flow provides arbitrarily complex posterior distribution, making it well suited for the inference network in variational autoencoder (Kingma and Welling, 2014). Inverse autoregressive flow (IAF) (Kingma et al., 2016) is a special type of normalizing flow where each invertible transformation is based on an autoregressive neural network. Thus, IAF can reuse the most successful autoregressive architecture, such as PixelCNN and WaveNet (van den Oord et al., 2016b;a). Learning an IAF with maximum likelihood can be very slow. In this work, we distill a Gaussian IAF from a pretrained autoregressive generative model by minimizing a numerically stable variant of KL divergence.

Knowledge distillation is originally proposed for compressing large models to smaller ones (Bucilua et al., 2006). In deep learning (Hinton et al., 2015), a smaller student network is distilled from the

teacher network by minimizing the loss between their outputs (e.g., L2 or cross-entropy). In parallel WaveNet, a non-autoregressvie student-net is distilled from an autoregressive WaveNet by minimizing the *reverse KL divergence* (Murphy, 2014). Similar techniques are applied in non-autoregressive models for machine translation (Gu et al., 2018; Kaiser et al., 2018; Lee et al., 2018; Roy et al., 2018).

## 3 Parallel Wave Generation

In this section, we present the Gaussian autoregressive WaveNet as the teacher-net and the Gaussian inverse autoregressive flow as the student-net. Then, we develop our knowledge distillation algorithm.

### 3.1 Gaussian Autoregressive WaveNet

WaveNet models the joint distribution of high dimensional waveform $\boldsymbol{x} = \{x_1, \ldots, x_T\}$ as the product of conditional distributions using the chain rules of probability,

$$p(\boldsymbol{x} \mid \boldsymbol{c} \,;\, \boldsymbol{\theta}) = \prod_{t=1}^{T} p(x_t \mid x_{<t}, \boldsymbol{c} \,;\, \boldsymbol{\theta}), \tag{1}$$

where $x_t$ is the $t$-th variable of $\boldsymbol{x}$, $x_{<t}$ represent all variables before $t$-step, $\boldsymbol{c}$ is the conditioner [2] (e.g., mel-spectrogram or hidden states in Section 4), and $\boldsymbol{\theta}$ are parameters of the model. The autoregressive WaveNet takes $x_{<t}$ as input, and outputs the probability distribution over $x_t$.

Parallel WaveNet (van den Oord et al., 2018) advocates mixture of logistics (MoL) distribution in PixelCNN++ (Salimans et al., 2017) for autoregressive teacher-net, as it requires much fewer output units compared to categorical distribution (e.g., 65,536 softmax units for 16-bit audios). Actually, the output distribution of student-net is required to be differentiable over samples $\boldsymbol{x}$ and allow backpropagation from teacher to student in distillation. As a result, one also needs to choose a continuous distribution for teacher WaveNet. Directly maximizing the log-likelihood of MoL is prone to numerical issues, and one has to employ the quantized surrogate loss introduced in PixelCNN++.

In this work, we demonstrate that a single Gaussian output distribution for WaveNet suffices to model the raw waveform. It might raise the modeling capacity concern because we use the single Gaussian instead of mixture of Gaussians (Chung et al., 2015). We will demonstrate their comparable performance in experiment. Specifically, the conditional distribution of $x_t$ given previous samples is,

$$p(x_t \mid x_{<t}; \boldsymbol{\theta}) = \mathcal{N}\big(\mu(x_{<t}; \boldsymbol{\theta}), \sigma(x_{<t}; \boldsymbol{\theta})\big), \tag{2}$$

where $\mu(x_{<t}; \boldsymbol{\theta})$ and $\sigma(x_{<t}; \boldsymbol{\theta})$ are mean and standard deviation predicted by the autoregressive WaveNet, respectively. In practice, the network predicts $\log \sigma(x_{<t})$ and operates at log-scale for numerical stability. Given observed data, we do maximum likelihood estimation (MLE) for $\boldsymbol{\theta}$. Note that, the model may give very accurate prediction of 16-bit discrete $x_t$ without real-valued noise (i.e., $\mu(x_{<t}) \approx x_t$), then the log-likelihood calculation can become numerically unstable when it is free to minimize $\sigma(x_{<t})$. As a result, we clip the predicted $\log \sigma(x_{<t})$ at $-9$ (natural logarithm) before calculating the log-likelihood at training. [3] We discuss the importance of clipping constant for log-scale in **Appendix A**. We also tried the dequantization trick by adding uniform noise $\boldsymbol{u} \in [0, \frac{2}{65536}]$ to the 16-bits samples similar as in image modeling (e.g., Uria et al., 2013). Indeed, these tricks are equivalent, in the sense that they both upper bound the continuous likelihood for modeling quantized data. We prefer the clipping trick, as it explicitly controls the model behavior and simplifies probability density distillation afterwards.

### 3.2 Gaussian Inverse Autoregressive Flow (IAF)

Normalizing flows (Rezende and Mohamed, 2015; Dinh et al., 2017) map a simple initial density $q(\boldsymbol{z})$ (e.g., isotropic Gaussian) into a complex one by applying an invertible transformation $\boldsymbol{x} = f(\boldsymbol{z})$. Given $f$ is a bijection, the distribution of $\boldsymbol{x}$ can be obtained through the change of variables formula:

$$q(\boldsymbol{x}) = q(\boldsymbol{z}) \left| \det\left( \frac{\partial f(\boldsymbol{z})}{\partial \boldsymbol{z}} \right) \right|^{-1}, \tag{3}$$

---

[2]We may omit $\boldsymbol{c}$ for concise notations.

[3]We clip log-scale at $-7$ at submission. Afterwards, we found smaller clipping constant (e.g., $-9$) gives better results for various datasets (e.g., Mandarin data), although it requires more iterations to converge.

where $\det\left(\frac{\partial f(\boldsymbol{z})}{\partial \boldsymbol{z}}\right)$ is the determinant of the Jacobian and is computationally expensive to obtain in general. Inverse autoregressive flow (IAF) (Kingma et al., 2016) is a special normalizing flow with a simple Jacobian determinant. In IAF, $\boldsymbol{z}$ has the same dimension as $\boldsymbol{x}$, and the transformation is based on an autoregressive network taking $\boldsymbol{z}$ as the input: $x_t = f(z_{\leq t}; \boldsymbol{\vartheta})$, where $\boldsymbol{\vartheta}$ are parameters of the model. Note that the $t$-th variable $x_t$ only depends on previous and current latent variables $z_{\leq t}$, thus the Jacobian is a triangular matrix and the determinant is the product of the diagonal entries,

$$\det\left(\frac{\partial f(\boldsymbol{z})}{\partial \boldsymbol{z}}\right) = \prod_t \frac{\partial f(z_{\leq t})}{\partial z_t}, \tag{4}$$

which is easy to calculate. Parallel WaveNet (van den Oord et al., 2018) uses a single logistic distribution based IAF to match its mixture of logistics (MoL) teacher.

We use the Gaussian IAF (Kingma et al., 2016) and define the transformation $x_t = f(z_{\leq t}; \boldsymbol{\vartheta})$ as:

$$x_t = z_t \cdot \sigma(z_{<t}; \boldsymbol{\vartheta}) + \mu(z_{<t}; \boldsymbol{\vartheta}), \tag{5}$$

where the shifting function $\mu(z_{<t}; \boldsymbol{\vartheta})$ and scaling function $\sigma(z_{<t}; \boldsymbol{\vartheta})$ are modeled by an autoregressive WaveNet in Section 3.1. The IAF transformation computes $\boldsymbol{x}$ in parallel given $\boldsymbol{z}$, which makes efficient use of resource like GPU. Importantly, if we assume $z_t \sim \mathcal{N}(z_t \mid \mu_0, \sigma_0)$, it is easy to observe that $x_t$ also follows a Gaussian distribution,

$$q(x_t \mid z_{<t}; \boldsymbol{\vartheta}) = \mathcal{N}(\mu_q, \sigma_q), \tag{6}$$

where $\mu_q = \mu_0 \cdot \sigma(z_{<t}; \boldsymbol{\vartheta}) + \mu(z_{<t}; \boldsymbol{\vartheta})$ and $\sigma_q = \sigma_0 \cdot \sigma(z_{<t}; \boldsymbol{\vartheta})$. Note that $\boldsymbol{x}$ are highly correlated through the marginalization of latents $\boldsymbol{z}$, and the IAF jointly models $\boldsymbol{x}$ at all timesteps.

To evaluate the likelihood of observed data $\boldsymbol{x}$, we can use the identities Eq. (3) and (4), and plug-in the transformation defined in Eq. (5), which will give us,

$$q(\boldsymbol{x}; \boldsymbol{\vartheta}) = q(\boldsymbol{z})\left(\prod_t \sigma(z_{<t}; \boldsymbol{\vartheta})\right)^{-1}. \tag{7}$$

However, one need the inverse transformationn $f^{-1}$ of Eq. (5),

$$z_t = \frac{x_t - \mu(z_{<t}; \boldsymbol{\vartheta})}{\sigma(z_{<t}, \vartheta)}, \tag{8}$$

to compute the corresponding $\boldsymbol{z}$ from the observed $\boldsymbol{x}$, which is autoregressive and slow. As a result, learning an IAF directly through maximum likelihood can be very slow.

In general, normalizing flows require a series of transformations until the distribution $q(\boldsymbol{x}; \boldsymbol{\vartheta})$ reaches a desired level of complexity. First, we draw a white noise sample $\boldsymbol{z}^{(0)}$ from the isotropic Gaussian distribution $\mathcal{N}(0, I)$. Then, we repeatedly apply the transformation $z_t^{(i)} = f(z_{\leq t}^{(i-1)}; \boldsymbol{\vartheta})$ defined in Eq. (5) from $\boldsymbol{z}^{(0)} \rightarrow \ldots \boldsymbol{z}^{(i)} \rightarrow \ldots \boldsymbol{z}^{(n)}$ and we let $\boldsymbol{x} = \boldsymbol{z}^{(n)}$. We summarize this procedure in Algorithm 1. Note the parameters are not shared across different flows.

## 3.3 KNOWLEDGE DISTILLATION

### 3.3.1 REGULARIZED KL DIVERGENCE

van den Oord et al. (2018) proposed the probability density distillation method to circumvent the difficulty of maximum likelihood learning for IAF. In distillation, the goal is to minimize the sequence-level reverse KL divergence between the student IAF and pretrained teacher WaveNet. This sequence-level KL divergence can be naively approximated by sampling $\boldsymbol{z}$ and $\boldsymbol{x} = f(\boldsymbol{z})$ from IAF, but it may exhibit high variance. The variance of this estimate can be reduced by marginalizing over the one-step-ahead predictions for each timestep (van den Oord et al., 2018). However, parallel WaveNet has to run a separate Monte Carlo sampling at each timestep, because the per-time-step KL divergence between the logistic and mixture of logistics distribution is still intractable. Indeed, parallel WaveNet first draws a white noise sample $\boldsymbol{z}$, then it draws multiple different samples $x_t$ from $q(x_t|z_{<t})$ to estimate the intractable integral. Our method only need to draw one sample $\boldsymbol{z}$, then it computes the KL divergence in closed-form thanks to the Gaussian setup.

---

**Algorithm 1** Gaussian Inverse Autoregressive Flows as Student Network

---

**Input:**     $z^{(0)} \sim \mathcal{N}(0, I)$: white noises;

             $n$: number of flows;

             $\{\vartheta^{(i)}\}$: parameters of autoregressive WaveNet for the $i$-th flow;

**Output:**  samples $x$;

             output distribution $q(x_t \mid z_{<t})$ with mean $\boldsymbol{\mu}_q[t]$ and standard deviation $\boldsymbol{\sigma}_q[t]$

Initialize $\boldsymbol{\mu}_z = 0, \;\; \boldsymbol{\sigma}_z = 1$

**for** $i$-th flow in $[1:n]$ **do**

    Run autoregressive WaveNet $\vartheta^{(i)}$ by taking $z^{(i-1)}$ as input

        $\boldsymbol{\mu}[t] \leftarrow \mu(z_{<t}^{(i-1)}; \vartheta^{(i)})$

        $\boldsymbol{\sigma}[t] \leftarrow \sigma(z_{<t}^{(i-1)}; \vartheta^{(i)})$

    $z^{(i)} = z^{(i-1)} \odot \boldsymbol{\sigma} + \boldsymbol{\mu}$

    $\boldsymbol{\sigma}_z = \boldsymbol{\sigma}_z \odot \boldsymbol{\sigma}$

    $\boldsymbol{\mu}_z = \boldsymbol{\mu}_z \odot \boldsymbol{\sigma} + \boldsymbol{\mu}$

**end for**

$x = z^{(n)}, \;\; \boldsymbol{\mu}_q = \boldsymbol{\mu}_z, \;\; \boldsymbol{\sigma}_q = \boldsymbol{\sigma}_z$

*Remark*: iterating over $\log \boldsymbol{\sigma}$ in log-scale improves numerical stability in practice.

---

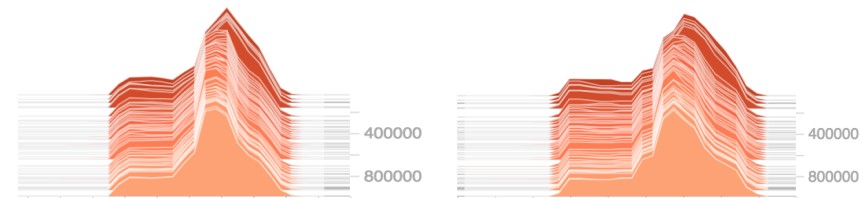

Figure 1: The empirical histograms of (a) $\log \sigma_p$ in teacher WaveNet and (b) $\log \sigma_q$ in student IAF during density distillation using reverse $\text{KL}^{\text{reg}}$ divergence.

Given a white noise sample $z$, Algorithm 1 outputs sample $x = f(z)$, as well as the output Gaussian distribution $q(x_t \mid z_{<t}; \vartheta)$ with mean $\mu_q$ and standard deviation $\sigma_q$. We feed the sample $x$ into an autoregressive WaveNet, and obtain its output distribution $p(x_t \mid x_{<t}; \theta)$ with mean $\mu_p$ and standard deviation $\sigma_p$. One can show that the per-time-step KL divergence between student's output distribution $q(x_t|z_{<t}; \vartheta)$ and teacher's $p(x_t|x_{<t}; \theta)$ has closed-form expression (see Appendix D),

$$\text{KL}\left(q \parallel p\right) = \log \frac{\sigma_p}{\sigma_q} + \frac{\sigma_q^2 - \sigma_p^2 + (\mu_p - \mu_q)^2}{2\sigma_p^2}, \tag{9}$$

which also forms an unbiased estimate of the sequence-level KL divergence between student's distribution $q(x)$ and teacher $p(x)$ (see Appendix C).

In this submission, we lower bound $\log \sigma_p$ and $\log \sigma_q$ at $-7$ before calculating the KL divergence. [4] However, the division by $\sigma_p^2$ still raises serious numerical problem, when we directly minimize the average KL divergence over all timesteps. To elaborate this, we monitor the empirical histograms of $\sigma_p$ from teacher WaveNet during distillation in Figure 1 (a). One can see that it is mostly distributed around $(e^{-9}, e^{-2})$, which incurs numerical problem if $\sigma_p$ and $\sigma_q$ have very different magnitudes at the beginning of training. This is because a well-trained WaveNet usually has highly peaked output distributions. The same observation holds true for other output distributions, including mixture of Gaussians and mixture of logistics.

To address this problem, we define the following variant of KL divergence:

$$\text{KL}^{\text{reg}}\left(q \parallel p\right) = \lambda \left| \log \sigma_p - \log \sigma_q \right|^2 + \text{KL}\left(q \parallel p\right). \tag{10}$$

---

[4]Clipping at $-6$ also works well and could improve numerical stability. See more discussion in Appendix A.

One can interpret the first term as regularization,[5] which largely stabilizes the optimization process by quickly matching the $\sigma$'s from student and teacher models, as demonstrated in Figure 1 (a) and (b). In addition, it does not introduce any bias for matching their probability density functions, as we have the following proposition:

**Proposition 3.1.** *For probability distributions in the location-scale family (including Gaussian, logistic distribution etc.), the regularized KL divergence in Eq.* (10) *still satisfies the following properties: (i)* $\text{KL}^{reg}\left(q \parallel p\right) \geq 0$, *and (ii)* $\text{KL}^{reg}\left(q \parallel p\right) = 0$ *if and only if* $p = q$.

Given a sample $z$ and its mapped $x$, we also test the *forward KL divergence* between the student's output distribution $q(x_t|z_{<t}; \boldsymbol{\vartheta})$ and teacher's $p(x_t|x_{<t}; \boldsymbol{\theta})$,

$$\text{KL}\left(p \parallel q\right) = \mathbb{H}(p, q) - \mathbb{H}(p), \tag{11}$$

where $\mathbb{H}(p, q)$ is the cross entropy, and $\mathbb{H}(p)$ is the entropy of teacher model. One can ignore the entropy term $\mathbb{H}(p)$ since we are optimizing student $q$ under a pretrained teacher $p$. Note that the sample-level forward KLD in Eq. (11) is a biased estimate of sequence-level KL $\left(p(\boldsymbol{x}) \parallel q(\boldsymbol{x})\right)$. To make it numerically stable, we apply the same regularization term in Eq. (10) and observe very similar empirical distributions of $\log \sigma$ in Figure 1.

### 3.3.2 STFT Loss

In knowledge distillation, it is a common practice to incorporate an additional loss using the ground-truth dataset (e.g., Kim and Rush, 2016). Empirically, we found that training student IAF with KL divergence loss alone will lead to whisper voices. van den Oord et al. (2018) advocates the *average* power loss to solve this issue, which is actually coupled with the short length of training audio clip (i.e. $0.32s$) in their experiments. As the clip length increases, the average power loss will be less effective. Instead, we compute the frame-level loss between the output samples $\boldsymbol{x}$ from student IAF and corresponding ground-truth audio $\boldsymbol{x}_n$:

$$\frac{1}{B}\left\| \left|\text{STFT}(\boldsymbol{x})\right| - \left|\text{STFT}(\boldsymbol{x}_n)\right| \right\|_2^2,$$

where $\left|\text{STFT}(\boldsymbol{x})\right|$ are the magnitudes of short-term Fourier transform (STFT), and $B = 1025$ is the number of frequency bins as we set FFT size to $2048$. We use a 12.5ms frame-shift, 50ms window length and Hanning window. Our final loss function is a linear combination of average KL divergence and frame-level loss, and we simply set their coefficients to one in all experiments.

## 4 TEXT-TO-WAVE ARCHITECTURE

In this section, we present our fully convolutional text-to-wave architecture (see Fig. 2 (a)) for end-to-end TTS. Our architecture is based on Deep Voice 3 (DV3), a convolutional attention-based TTS system (Ping et al., 2018). DV3 is capable of converting textual features (e.g., characters, phonemes and stresses) into spectral features (e.g., log-mel spectrograms and log-linear spectrograms). These spectral features can be used as inputs for a separately trained waveform synthesis model, such as WaveNet. In contrast, we directly feed the hidden representation learned from the attention mechanism to the WaveNet through some intermediate processing, and train the whole model from scratch in an end-to-end manner.

Note that, conditioning the WaveNet on hidden representation is crucial to the success of training from scratch. Indeed, we tried to condition WaveNet on predicted mel-spectrogram from DV3, thus the gradients of WaveNet loss can backpropagate through DV3 to improve the text-to-spectrogram model. When the whole model is trained from scratch, we found it performs slightly worse than the separate training pipeline. The major reason is that the predicted mel-spectrogram from DV3 can be inaccurate at early training, and may spoil the training of WaveNet. In order to get satisfactory results, one need pretrain DV3 and WaveNet, then fine-tune the whole system (e.g., Zhao et al., 2018).

The proposed architecture consists of four components:

---

[5]We fix $\lambda = 4$ in all experiments.

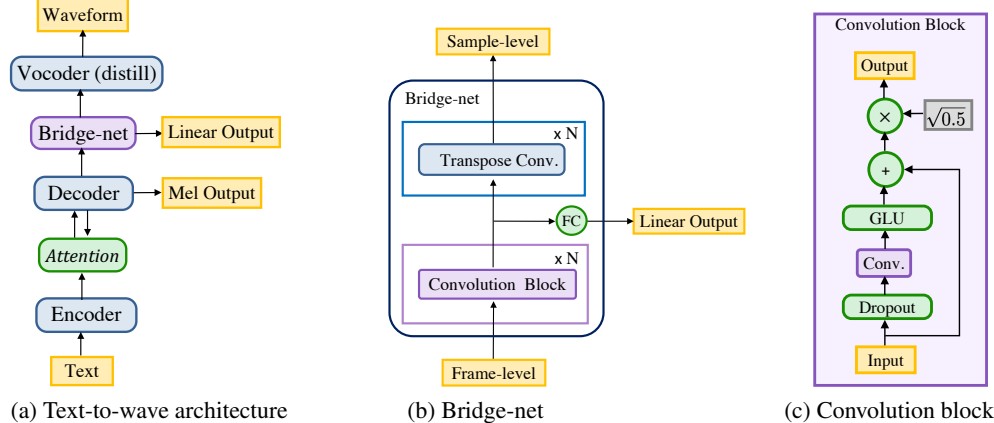

(a) Text-to-wave architecture      (b) Bridge-net      (c) Convolution block

Figure 2: (a) Text-to-wave model converts textual features into waveform. All components feed their hidden representation to others directly. (b) Bridge-net maps frame-level hidden representation to sample-level through several convolution blocks and transposed convolution layers interleaved with *softsign* non-linearities. (c) Convolution block is based on gated linear unit.

- **Encoder**: A convolutional encoder as in DV3, which encodes textual features into an internal hidden representation.
- **Decoder**: A causal convolutional decoder as in DV3, which decodes the encoder representation with attention into the log-mel spectrogram in an autoregressive manner.
- **Bridge-net**: A convolutional intermediate processing block, which processes the hidden representation from the decoder and predict log-linear spectrogram. Unlike the decoder, it is non-causal and can thus utilize future context information. In addition, it upsamples the hidden representation from frame-level to sample-level.
- **Vocoder**: A Gaussian autoregressive WaveNet to synthesize the waveform, which is conditioned on the upsampled hidden representation from the bridge-net. This component can be replaced by a student IAF distilled from the autoregressive vocoder.

The overall objective function is a linear combination of the losses from decoder, bridge-net and vocoder; we simply set all coefficients to one in experiments. We introduce bridge-net to utilize future temporal information as it can apply non-causal convolution. All modules in our architecture are convolutional, which enables fast training [6] and alleviates the common difficulties in RNN-based models (e.g., vanishing and exploding gradient problems (Pascanu et al., 2013)). Throughout the whole model, we use the *convolution block* from DV3 (see Fig. 2(c)) as the basic building block. It consists of a 1-D convolution with a gated linear unit (GLU) and a residual connection. We set the dropout probability to 0.05 in all experiments. We give further details in the following subsections.

### 4.1 ENCODER-DECODER

We use the same encoder-decoder architecture as DV3 (Ping et al., 2018). The encoder first converts characters or phonemes into trainable embeddings, followed by a series of convolution blocks to extract long-range textual information. The decoder autoregressively predicts the log-mel spectrograms with an L1 loss (teacher-forced at training). It starts with layers of 1x1 convolution to preprocess the input log-mel spectrogram, and then applies a series of causal convolutions and attentions. A multi-hop attention-based alignment is learned between character embeddings and log-mel spectrograms.

### 4.2 BRIDGE-NET

The hidden states of decoder are fed to the bridge-net for temporal processing and upsampling. The output hidden representation is then fed to the vocoder for waveform synthesis. Bridge-net consists of a stack of convolution blocks, and two layers of transposed 2-D convolution interleaved with *softsign*

---

[6]For example, DV3 trains an order of magnitude faster than its RNN peers.

| Output Distribution | Subjective 5-scale MOS | Test CLL |
|---|:---:|:---:|
| Gaussian | $4.40 \pm 0.20$ | 4.687 |
| Mixture of Gaussians | $4.38 \pm 0.22$ | 4.671 |
| Mixture of Logistics | $4.03 \pm 0.27$ | 4.590 |
| Softmax (2048-way) | $4.31 \pm 0.23$ | — |
| Ground-truth (24 kHz) | $4.54 \pm 0.12$ | — |

Table 1: Mean Opinion Score (MOS) ratings with 95% confidence intervals using different output distributions for autoregressive WaveNet. We also include the conditional log-likelihoods (CLL) (per dimension) on the same 16 test audios for WaveNet with continuous outputs.

| Distillation method | Subjective 5-scale MOS |
|---|:---:|
| Student-1 with reverse KL$^{\text{reg}}$ | $4.16 \pm 0.21$ |
| Student-1 with forward KL$^{\text{reg}}$ | $4.12 \pm 0.20$ |
| Student-2 with reverse KL$^{\text{reg}}$ | $4.22 \pm 0.17$ |

Table 2: Mean Opinion Score (MOS) ratings with 95% confidence intervals using different distillation objective functions for student Gaussian IAF. We use the crowdMOS toolkit as in Table 1.

to upsample the per-timestep hidden representation from 80 per second to 24,000 per second. The upsampling strides in time are 15 and 20 for the two layers, respectively. Correspondingly, we set the 2-D convolution filter sizes as $(30, 3)$ and $(40, 3)$, where the filter sizes (in time) are doubled from strides to avoid the checkerboard artifacts (Odena et al., 2016).

## 5 EXPERIMENT

In this section, we present several experiments to evaluate the proposed parallel wave generation method and text-to-wave architecture.

**Data:** We use an internal English speech dataset containing about 20 hours of audio from a female speaker with a sampling rate of 48 kHz. We downsample the audios to 24 kHz.

**Autoregressive WaveNet:** We first show that a single Gaussian output distribution for autoregressive WaveNet suffices to model the raw waveform. We use the similar WaveNet architecture detailed in Arık et al. (2017a) (see Appendix B). We use 80-band log-mel spectrogram as the conditioner. To upsample the conditioner from frame-level (80 per second) to sample-level (24,000 per second), we apply two layers of transposed 2-D convolution (in time and frequency) interleaved with leaky ReLU ($\alpha = 0.4$). The upsampling strides in time are 15 and 20 for the two layers, respectively. Correspondingly, we set the 2-D convolution filter sizes as $(30, 3)$ and $(40, 3)$. We also find that normalizing log-mel spectrogram to the range of [0, 1] improves the synthesized audio quality (e.g., Yamamoto, 2018). We train 20-layers WaveNets conditioned on ground-truth log-mel spectrogram with various output distributions, including single Gaussian, 10-component mixture of Gaussians (MoG), 10-component mixture of Logistics (MoL), and softmax with 2048 linearly quantized channels. We set both residual channel (dimension of the hidden state of every layer) and skip channel (the dimension to which layer outputs are projected prior to the output layer) to 128. We set the filter size of dilated convolutions to 2 for teacher WaveNet. All models share the same architecture except the output distributions, and they are trained for 1000K steps using the Adam optimizer (Kingma and Ba, 2015) with batch-size 8 and 0.5s audio clips. The learning rate is set to 0.001 in the beginning and annealed by half for every 200K steps.

We report the mean opinion score (MOS) for naturalness evaluation in Table 1. We use the crowdMOS toolkit (Ribeiro et al., 2011), where batches of samples from these models were presented to workers on Mechanical Turk. The results indicate that the Gaussian autoregressive WaveNet provides comparable results to MoG and softmax outputs, and outperforms MoL in our experiments. We also include the conditional log-likelihoods (CLL) on test audios (conditioned on mel-spectrograms) for continuous output WaveNets, where the Gaussian, MoG, and MoL are trained with the same clipping constant $-9$. See more discussions about clipping constant for log-scale variable in Appendix A. MoL obtains slightly worse CLL, as it does not directly optimize the continuous likelihood.

**Student Gaussian IAF:** We distill two 60-layer parallel student-nets from a pre-trained 20-layer Gaussian autoregressive WaveNet. Our student-1 consists six stacked Gaussian IAF and each flow

| Method | Subjective 5-scale MOS |
|---|---|
| Text-to-Wave Model | $4.15 \pm 0.25$ |
| Text-to-Wave (distilled vocoder) | $4.11 \pm 0.24$ |
| DV3 + WaveNet (predicted Mel) | $3.81 \pm 0.26$ |
| DV3 + WaveNet (true Mel) | $3.73 \pm 0.24$ |

Table 3: Mean Opinion Score (MOS) ratings with 95% confidence intervals for comparing the text-to-wave model and separately trained pipeline. We use the crowdMOS toolkit as in Table 1.

is parameterized by a 10-layer WaveNet with 64 residual channels, 64 skip channels, and filter size 3 in dilated convolutions. Student-2 consists of four stacked Gaussian IAF blocks, which are parameterized by [10, 10, 10, 30]-layer WaveNets respectively, with the same channels and filter size as studuent-1. For student-2, we also reverse the sequence being generated in time between successive IAF blocks and find it improves the performance. Note that the student models share the same conditioner network (layers of transposed 2-D convolution) with teacher WaveNet during distillation. Training conditioner network of student model from scratch leads to worse result. We test both the forward and reverse KL divergences combined with the STFT-loss, and we simply set their combination coefficients to one in all experiments. The student models are trained for 1000K steps using Adam optimizer. The learning rate is set to 0.001 in the beginning and annealed by half for every 200K steps. Surprisingly, we always find good results after only 50K steps of distillation, which perhaps benefits from the closed-form computation of KL divergence. The models are trained longer for extra improvement. We report the MOS evaluation results in Table 2. Both of these distillation methods work well and obtain comparable results. Student-2 outperforms student-1 by generating "clearner" voices. We expect further improvements by incorporating perceptual and contrastive losses introduced in van den Oord et al. (2018) and we will leave it for future work. At inference, the parallel student-net runs ∼20 times faster than real time on NVIDIA GeForce GTX 1080 Ti.

**Text-to-Wave Model:** We train the proposed text-to-wave model from scratch and compare it with the separately trained pipeline presented in Deep Voice 3 (DV3) (Ping et al., 2018). We use the same text preprocesssing and joint character-phoneme representation in DV3. The hyper-parameters of encoder and decoder are the same as the single-speaker DV3. The bridge-net has 6 layers of convolution blocks with input/output size of 256. The hyper-parameters of the vocoders are the same as previous subsections. The vocoder part is trained by conditioning on sliced hidden representations corresponding to 0.5s audio clips. Other parts of model are trained on whole-length utterances. The model is trained for 1.5M steps using Adam optimizer with batch size 16. The learning rate is set to 0.001 in the beginning and annealed by half for every 500K steps. We also distill a Gaussian IAF from the trained autoregressive vocoder within this end-to-end model. Both student IAF and autoregressive vocoder are conditioned on the upsampled hidden representation from the bridge-net. For the separately trained pipeline, we train two Gaussian autoregressive WaveNets conditioned on ground-truth mel-spectrogram and predicted mel-spectrogram from DV3, respectively. We run inference on the same unseen text as DV3 and report the MOS results in Table 3. The results demonstrate that the text-to-wave model significantly outperforms the separately trained pipeline. The text-to-wave model with a distilled parallel vocoder gives slightly worse result to the one with autoregressive vocoder. In the separately trained pipeline, training a WaveNet conditioned on predicted mel-spectrograms eases the training/test mismatch, thus outperforms training with ground-truth.

## 6 CONCLUSION

In this work, we first demonstrate that a single Gaussian output distribution is sufficient for modeling the raw waveform in WaveNet without degeneration of audio quality. Then, we propose a parallel wave generation method based on Gaussian inverse autoregressive flow (IAF), in which the IAF is distilled from the autoregressive WaveNet by minimizing a regularized KL divergence for highly peaked distributions. In contrast to parallel WaveNet, our distillation algorithm estimates the KL divergence in closed-form and largely stabilizes the training procedure. Furthermore, we propose the first text-to-wave neural architecture for TTS, which can be trained from scratch in an end-to-end manner. Our text-to-wave architecture outperforms the separately trained pipeline and opens up the research opportunities for fully end-to-end TTS. We also demonstrate appealing results by distilling a parallel neural vocoder conditioned on the hidden representation within the end-to-end model.

ACKNOWLEDGEMENTS

We thank Yongguo Kang, Yu Gu and Tao Sun from Baidu Speech Department for very helpful discussions. We also thank anonymous reviewers for their valuable feedback and suggestions.

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

# Appendices

## A CLIPPING LOG-SCALE AT TRAINING

Clipping for $\log \sigma$ plays an important role in training Gaussian WaveNet. Without the clipping trick, the optimization process become numerically unstable. The clipping constant also controls the model capacity and largely impacts on final speech quality. We discuss its impact for both autoregressive WaveNet and student IAF. Note that, the clipping is only applied at training, not at inference.

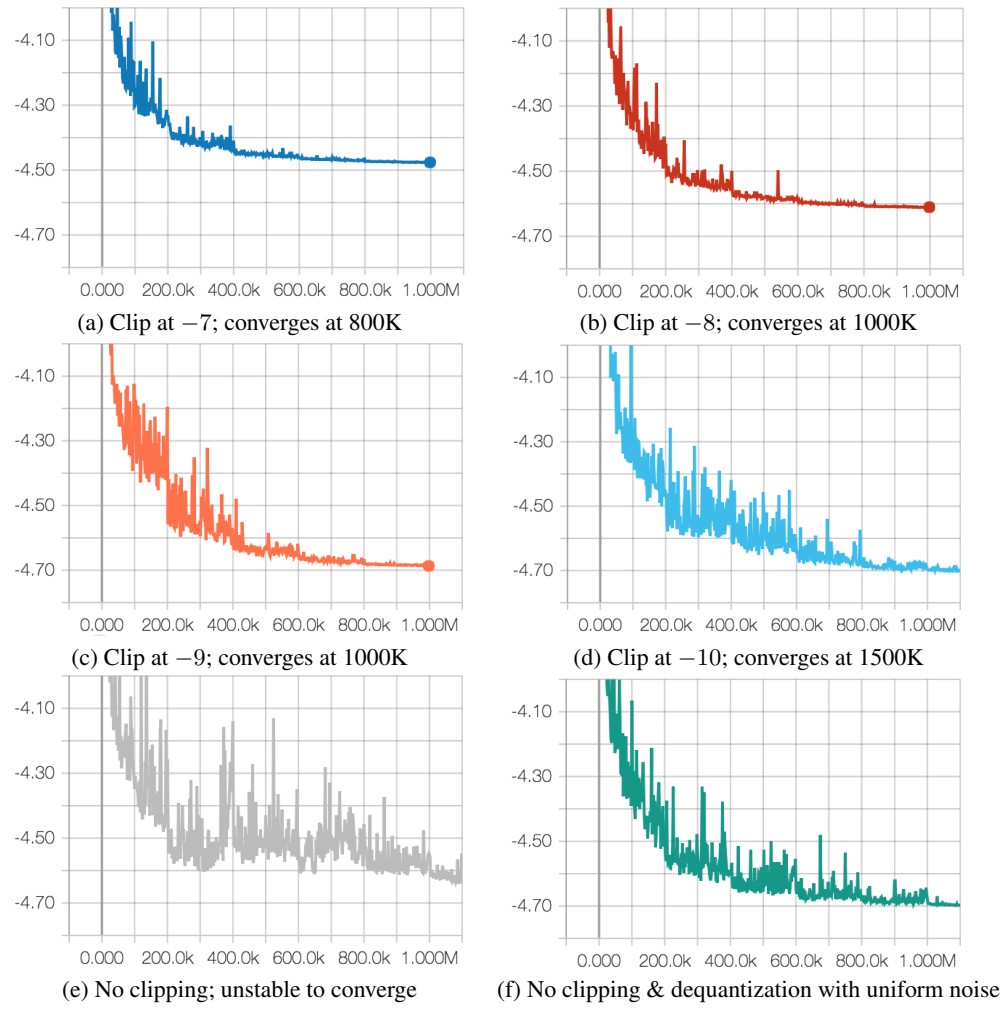

Figure 3: The negative log-likelihoods (per dimension) of Gausssian WaveNet on hold-out audios during training. The learning rates in Adam optimizer are initially set to 0.001 and annealed by half for every 200K steps.

## A.1 AUTOREGRESSIVE WAVENET

For Gaussian WaveNet alone, **smaller clipping** constant for $\log \sigma(x_{<t})$ at training usually leads to **larger likelihood**, but it also need **more iterations** to converge. Figure 3 shows its impact on log-likelihood and convergence behaviour. From Figure 3 (a)-(d), the validation likelihood improves a lot from with clipping constant $-7$ to $-9$, but the improvement is negligible from $-9$ to $-10$. In addition, (e) shows the numerical instability without clipping, and (f) shows the dequantization with uniform noise $\boldsymbol{u} \in [0, \frac{2}{65536}]$ stabilizes optimization and performs very similar as clipping at $-10$.

For speech quality, models trained with small clipping constant (e.g., $-9$) tend to have less artifacts at convergence, especially for the silence portion of utterances. However, very small clipping constant in teacher WaveNet may raise difficulty for distillation, because the range of $\log \sigma$ will be large (see Figure 4). For different datasets and conditioners, the optimal clipping constant may be different. **We suggest $-9$ as the default for Gaussian teacher**, after we tried various datasets (including English, Mandarin) and conditioners (including mel-spectrogram, hidden states, linguistic conditioner).

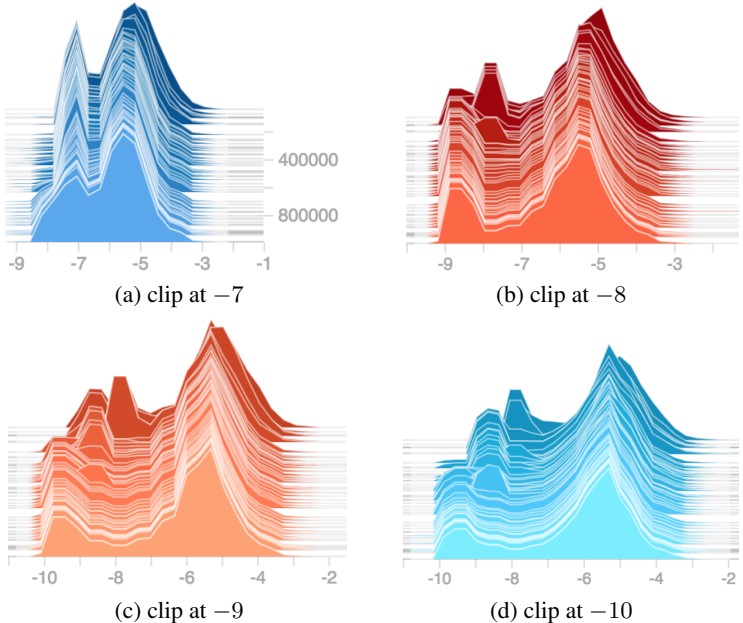

Figure 4: The empirical histograms of predicted $\log \sigma$ (before clipping) in Gaussian WaveNet with different clipping constants during training.

### A.2 GAUSSIAN IAF

In distillation, we also clip $\log \sigma_p$ and $\log \sigma_q$ for numerical reason before computing the KL divergence (KLD). Note that, the clipping is not applied for the regularization term. In general, larger clipping constant leads to more stable optimization, but it could make the KLD loss less useful. **We suggest $-6$ as the default setting**, after we tried various datasets and conditioners.

**Useful tricks:** When we work on student WaveNet with linguistic conditioner on internal Mandarin dataset, we find the following tricks are effective to improve the numerical stability at distillation.

- After initial training (e.g., 500 iterations), if the KLD loss is larger than a threshold (e.g., 10.0), we simply mask it as zero, and let the regularization term and STFT loss to help it out.
- After initial training, if the global norm of gradients is larger than 1000.0, we clip the gradients by small values $[-0.1, 0.1]$. Otherwise, we clip the values of gradients to $[-5.0, 5.0]$.
- Larger batch size (e.g., 16) and smaller learning rate (e.g., 0.0002) are helpful to stabilize the distillation.

## B DETAILS OF DILATED CONVOLUTION BLOCK

We also employ a stack of dilated convolution blocks, where each block has 10 layers and the dilation is doubled at each layer, i.e., $\{1, 2, 4, ..., 512\}$. We add the output hidden states from each layer through residual connection before projecting them to the number of skip channels.

In dilated convolution block, we compute the $i$-th hidden layer $\boldsymbol{h}^{(i)}$ with dilation $2^{i-1}$ by gated convolutions (van den Oord et al., 2016b):

$$\boldsymbol{h}^{(i)} = \text{sigmoid}(W_g^{(i)} * \boldsymbol{h}^{(i-1)} + A_g^{(i)} \cdot \boldsymbol{c} + b_g^{(i)}) \odot \tanh(W_f^{(i)} * \boldsymbol{h}^{(i-1)} + A_f^{(i)} \cdot \boldsymbol{c} + b_f^{(i)}),$$

therein $h^0 = x$ is the input of the block, $*$ denotes the causal dilated convolution, $\cdot$ represents $1 \times 1$ convolution over the *upsampled* conditioner $c$, $\odot$ denotes the element-wise multiplication, $W_g^{(i)}, A_g^{(i)}, b_g^{(i)}$ are convolutions and bias parameters at $i$-th layer for *sigmoid* gating function, and $W_f^{(i)}, A_f^{(i)}, b_f^{(i)}$ are analogous parameters for *tanh* function.

## C  ESTIMATE THE SEQUENCE-LEVEL KL DIVERGENCE

The sequence-level KL divergence between student distribution $q(x)$ and teacher's $p(x)$ can be written as,

$$
\begin{aligned}
\text{KL}\left(q(x) \parallel p(x)\right) &= \mathbb{E}_{q(x)}\left[\log q(x) - \log p(x)\right], \\
&\text{note that, } q(x) = \prod_{t=1}^{T} q(x_t|x_{<t}) \text{ and } p(x) = \prod_{t=1}^{T} p(x_t|x_{<t}), \\
&= \mathbb{E}_{q(x)}\left[\sum_{t=1}^{T} \log q(x_t|x_{<t}) - \log p(x_t|x_{<t})\right] \\
&= \sum_{t=1}^{T} \mathbb{E}_{q(x_{\leq t})}\left[\log q(x_t|x_{<t}) - \log p(x_t|x_{<t})\right] \\
&= \sum_{t=1}^{T} \mathbb{E}_{q(x_{<t})}\,\mathbb{E}_{q(x_t|x_{<t})}\left[\log q(x_t|x_{<t}) - \log p(x_t|x_{<t})\right] \\
&= \sum_{t=1}^{T} \mathbb{E}_{q(x_{<t})}\left[\text{KL}\left(q(x_t|x_{<t}) \parallel p(x_t|x_{<t})\right)\right] \\
&= \mathbb{E}_{q(x)} \sum_{t=1}^{T}\left[\text{KL}\left(q(x_t|x_{<t}) \parallel p(x_t|x_{<t})\right)\right]
\end{aligned}
$$

Note that, the above equality holds for arbitrary distributions. Since $q(x)$ is an IAF, and $x$ are sampled through $x = f(z)$ and $z \sim N(0, I)$, then

$$
\text{KL}\left(q(x) \parallel p(x)\right) = \mathop{\mathbb{E}}_{\substack{z \sim N(0,I) \\ x = f(z)}}\left[\sum_{t=1}^{T} \text{KL}\left(q(x_t|z_{<t}) \parallel p(x_t|x_{<t})\right)\right].
$$

Thus, the summation of per-time-step KL divergence is an unbiased estimate of the sequence-level KL divergence.

## D  KL DIVERGENCE BETWEEN GAUSSIAN DISTRIBUTIONS

Given two Gaussian distributions $p(x) = \mathcal{N}(\mu_p, \sigma_p)$ and $q(x) = \mathcal{N}(\mu_q, \sigma_q)$, their KL divergence is:

$$
\text{KL}\left(q \parallel p\right) = \int q(x) \log \frac{q(x)}{p(x)} dx = \mathbb{H}(q, p) - \mathbb{H}(q)
$$

where $\log \equiv \log_e$, the entropy,

$$
\begin{aligned}
\mathbb{H}(q) &= -\int q(x) \log q(x) dx \\
&= -\int q(x) \log \left[ (2\pi\sigma_q^2)^{-\frac{1}{2}} \exp\left(-\frac{(x-\mu_q)^2}{2\sigma_q^2}\right) \right] dx \\
&= \frac{1}{2} \log\left(2\pi\sigma_q^2\right) \int q(x) dx \ + \ \frac{1}{2\sigma_q^2} \int q(x)(x-\mu_q)^2 dx \\
&= \frac{1}{2} \log\left(2\pi\sigma_q^2\right) \cdot 1 \ + \ \frac{1}{2\sigma_q^2} \cdot \sigma_q^2 \\
&= \frac{1}{2} \log\left(2\pi\sigma_q^2\right) + \frac{1}{2}
\end{aligned}
$$

and the cross entropy,

$$
\begin{aligned}
\mathbb{H}(q, p) &= -\int q(x) \log p(x) dx \\
&= -\int q(x) \log \left[ (2\pi\sigma_p^2)^{-\frac{1}{2}} \exp\left(-\frac{(x-\mu_p)^2}{2\sigma_p^2}\right) \right] dx \\
&= \frac{1}{2} \log\left(2\pi\sigma_p^2\right) \int q(x) dx \ + \ \frac{1}{2\sigma_p^2} \int q(x)(x-\mu_p)^2 dx \\
&= \frac{1}{2} \log\left(2\pi\sigma_p^2\right) \ + \ \frac{1}{2\sigma_p^2} \int q(x)(x^2 - 2\mu_p x + \mu_p^2) dx \\
&= \frac{1}{2} \log\left(2\pi\sigma_p^2\right) \ + \ \frac{\mu_q^2 + \sigma_q^2 - 2\mu_p\mu_q + \mu_p^2}{2\sigma_p^2} \\
&= \frac{1}{2} \log\left(2\pi\sigma_p^2\right) \ + \ \frac{\sigma_q^2 + (\mu_p - \mu_q)^2}{2\sigma_p^2}.
\end{aligned}
$$

Combining $\mathbb{H}(q)$ and $\mathbb{H}(q, p)$ together, we obtain

$$
\mathrm{KL}\left(q \parallel p\right) = \log\frac{\sigma_p}{\sigma_q} + \frac{\sigma_q^2 - \sigma_p^2 + (\mu_p - \mu_q)^2}{2\sigma_p^2}.
$$

