# OpenReview forum: "ClariNet: Parallel Wave Generation in End-to-End Text-to-Speech"
_ICLR.cc/2019/Conference_

### Official Review · AnonReviewer2 · 2018-10-21
**Good and significant work, paper could be improved**

**Rating:** 7
**Confidence:** 4

**Review:**

Paper summary:

The paper presents two distinct contributions in text-to-speech systems:
a) It describes a method for distilling a Gaussian WaveNet into a Gaussian Inverse Autoregressive Flow that uses an analytically computed KL between their conditionals.
b) It presents a text-to-speech system that is trained end-to-end from text to waveforms.

Technical quality:

The distillation method presented in the paper is technically correct. The evaluation is based on Mean Opinion Score and seems to follow good practices.

The paper makes three claims:
a) A WaveNet with Gaussian conditionals can model speech waveforms equally well as WaveNets with other types of conditionals.
b) Analytically computing KL divergence stabilizes distillation.
c) A text-to-speech system trained end-to-end from text to waveforms outperforms one that has separately trained text-to-spectrogram and spectrogram-to-waveform subsystems.

Claims (a) and (c) are clearly demonstrated in the experiments. However, there is nothing in the paper that substantiates claim (b). I think the paper would be strengthened if the performance of sample-based KL distillation was added into Table 2, and if learning curves were reported that evaluate the amount of stabilization that an analytical KL may offer vs a sample-based KL.

Further points about the experiments:
- It wasn't clear to me whether distillation happens at the same time as the autoregressive WaveNet is trained on data, or after it has been fully trained. I think the paper should make this clear.
- The paper says that distillation makes generation three orders of magnitude faster. I think it would be good if actual generation times (e.g. in seconds) were reported.

Clarity:

The paper is generally well-written. Sections 1 and 2 in particular are excellent.

However, section 3 contains several notational errors and technical inaccuracies, that makes it rather confusing to read. In particular:
- q(x_t | z_{<=t}) is used in several places to mean the Gaussian conditional q(x_t | z_{<t}) (e.g. in Eqs (6) and (7), and elsewhere). This is confusing, as q(x_t | z_{<=t}) is actually a delta distribution.
- q(x | z) is used in several places to mean q(x) (e.g. in Eq. (7), in Alg. 1 and elsewhere). This is confusing, as q(x | z) is also a delta distribution.
I believe that section 3, especially subsections 3.2 and 3.3.1, should be reworked to be made clearer, and the notation should be carefully revised.

I don't think the paper needs to span 9 pages. Section 3 is rather wordy, and should be compressed to the important points.

Originality:

Distilling a Gaussian autoregressive model to another Gaussian autoregressive model by matching their Gaussian conditionals with an analytical KL is rather straightforward, and, methodologically speaking, I wouldn't consider it an original contribution on its own. However, I think its application and demonstration in text-to-speech constitutes an original contribution.

Significance:

The paper contains a substantial amount of significant work that I think is important to be communicated to the ICLR community, especially the text-to-speech community.

Review summary:

Pros:
+ Substantial amount of good work.
+ Significant improvement in text-to-speech end-to-end software.
+ Generally well-written (with the exception of section 3 which needs work).

Cons:
- Some more experiments would be good to substantiate the claim that analytical KL is better.
- Notational errors and confusion in section 3.
- Too wordy, no need for 9 pages.

Nitpicks:
- As I said above, I wouldn't consider distillation of models with Gaussian conditionals using analytical KLs methodologically novel, so I think the phrase "novel regularized KL divergence" should be moderated.
- Eq. (1) should contain theta on the left hand side too.
- Page 3: "at Appendix B" --> "in Appendix B".
- Page 4: In flows we don't just "suppose z has the same dimension as x"; rather, it's a necessary condition that must hold.
- Footnote 5: It's unclear to me what it means to "make the loss less sensitive".
- References: Real NVP, Fourier, Bayes, PixelCNN, WaveNet, VoiceLoop should be properly capitalized.

---

> ### Author Response · Authors · 2018-11-27
> **To Reviewer 2:**
>
>
> Thank you for your in-depth review. These comments are really helpful for improving our paper.
>
> - “I think the paper would be strengthened if the performance of sample-based KL distillation was added into Table 2, and if learning curves were reported that evaluate the amount of stabilization that an analytical KL may offer vs a sample-based KL.”
> * This is a good point. We tried to implement sample-based KL distillation with mixture of logistic distribution, but we haven’t produced high quality speech as provided by the original paper. We didn’t include the results in Table 2, as it seems like we are comparing with a straw man. It should be noted, at the time of this submission, implementing parallel WaveNet is still beyond the capabilities of open source community, even though it receives a lot of attention from TTS practitioners and is very valuable for TTS production. For example, there are some open discussion in the following repo:  https://github.com/r9y9/wavenet_vocoder/issues/7  .
> In addition, many thanks for your suggestion about learning curves. We will add the comparison of learning curves (analytical KL vs sample-based KL) in our final draft.
>
> - “It wasn't clear to me whether distillation happens at the same time as the autoregressive WaveNet is trained on data, or after it has been fully trained. I think the paper should make this clear.”
> * The distillation happens after the autoregressive WaveNet is fully trained. We have clarified this in our revision.
>
> - “However, section 3 contains several notational errors … q(x_t | z_{<=t}) is used in several places to mean the Gaussian conditional q(x_t | z_{<t}) … I believe that section 3, especially subsections 3.2 and 3.3.1, should be reworked to be made clearer, and the notation should be carefully revised.”
> * Many thanks for pointing it out. Yes, q(x_t | z_{<t}) is Gaussian, but q(x_t | z_{<=t}) is deterministic a.k.a. a delta distribution. The Eq. (7) is technically true, only because all the involved distributions q(z|x) and q(x_t | z_{<=t}) ) are deterministic (as pointed out by Reviewer 3).  To avoid confusion, we have removed Eq. (7) and related description. Also, we have revised these notation errors in Section 3.
>
> - “I don't think the paper needs to span 9 pages. Section 3 is rather wordy, and should be compressed to the important points.”
> * We have shortened Section 3 in our revision. We will further shorten it in our final draft.
>
> - “The paper contains a substantial amount of significant work that I think is important to be communicated to the ICLR community, especially the text-to-speech community.”
> * We really appreciate your comment.
>
> We have fixed the issues listed in “Nitpicks”. Many thanks for your detailed review.

---

> > ### Comment · AnonReviewer2 · 2018-11-28
> > **Thanks**
> >
> > Many thanks for your detailed response, and for addressing many of the issues.
> >
> > I remain positive about the paper, and I'd be very pleased if my review has helped in improving it. I think it's good work, and I wish it best of luck.

---

### Official Review · AnonReviewer1 · 2018-11-08
**A strong result but unclear experiments and contribution**

**Rating:** 6
**Confidence:** 3

**Review:**

After reading other reviews and author comments, I have raised my rating to a 6. My main concerns remain (lack of significant contribution and lack of an ablation study with more comprehensive experiments). However, I'm not against the paper as an interesting finding in and of itself. It would be great if the authors (or interested members of the research community) may analyze how general-purpose their proposals are (e.g., of Gaussian base distribution) and how extensive the results are on TTS benchmarks.

--

Overall, I very much like the direction this paper pursues. However, the content doesn't substantiate their two claimed contributions. I highly recommend the authors either back up their claims in more detail, or center their work in terms of the result and less so about the ideas (which at the moment, are not convincing to use outside of this specific setup).

The authors propose two contributions:

1. They build on parallel WaveNet which uses distillation by minimizing a KL divergence from a Logistic IAF as a student to a Mixture of Logistic AF as a teacher. Instead, they simply use Gaussians which has a closed-form KL divergence and makes training during distillation significantly simpler. Because of stability problems, they also add 1. a penalty term to discourage the original loss from dividing by a standard deviation close to zero; and 2. converting van den Oord et al. (2018)'s average power loss penalty to a frame-level loss penalty.

Their choice of Gaussians requires a restriction on the likelihood, and they show one result arguing the likelihood choice doesn't make much of a difference. This result comprises 4 human-evaluated numbers, with a fixed architecture and training hyperparameters of their choice. Unfortunately, I'm not convinced. Can the authors provide more compelling evidence? If the authors argue this is one of their main contributions, I find that lack of a more comprehensive empirical or theoretical study disconcerting.

Similarly, while I like that using Gaussian KLs makes the distillation objective in closed-form, there isn't evidence indicating the benefit. The one result (the 4 numbers above) are conflated by both the change in model as well as utilizing the closed-form loss. The same goes for their one result (2 numbers) comparing forward to reverse KL.

2. They "propose the first text-to-wave neural architecture for TTS, which can be trained from scratch in an end-to-end
manner." I'm not an expert on speech so I can't accurately assess the novelty here. However, it would be nice to show these results independent of the other proposed changes.

Writing-wise, the paper was clear, although potentially too packed with background information. As a expert on generative models, most of Sections 1-3 are already well-known and could be made more concise by referencing past works for more details. They add various details (such as the architecture notes at the end of 3.1) which should be better placed elsewhere to tease out what the important changes are in this paper.

---

> ### Author Response · Authors · 2018-11-26
> **To Reviewer 1:**
>
>
> Thank you for your review; the feedback is very helpful to improve our paper.
>
> - “Their choice of Gaussians requires a restriction on the likelihood, and they show one result arguing the likelihood choice doesn't make much of a difference. This result comprises 4 human-evaluated numbers, with a fixed architecture and training hyperparameters of their choice. Unfortunately, I'm not convinced. Can the authors provide more compelling evidence? If the authors argue this is one of their main contributions, I find that lack of a more comprehensive empirical or theoretical study disconcerting”
> * It is a standard practice to evaluate Mean Opinion Score (MOS) in speech synthesis. Although they are human-evaluated numbers from crowdsourcing platform (Mechanical Turk), they are much more indicative of the true goal (a.k.a. synthesizing high fidelity speech) than any other objective metrics, as long as the MOS evaluation follows good practices (e.g., crowdMOS). Also, we didn’t claim that autoregressive WaveNet with single Gaussian outperforms other options (e.g., softmax). Instead, we argue it is sufficient for modeling the raw waveform in WaveNet by providing competitive quality of synthesized speech as others options.
>
> - “Similarly, while I like that using Gaussian KLs makes the distillation objective in closed-form, there isn't evidence indicating the benefit."
> * This is a good point. One major benefit of Gaussian is the closed-form distillation objective, in contrast to parallel WaveNet. We tried to implement parallel WaveNet with Monte Carlos estimates of the KLD, but we haven’t produced high quality speech as provided by the original paper. We didn’t include this result, as it may impose the impression that we are comparing with a strawman. It should be noted, at the time of this submission, implementing parallel WaveNet is still beyond the capabilities of open source community, even though it attracts a lot of attention from TTS practitioners and is very valuable for TTS production. For example, here is some open discussion ( https://github.com/r9y9/wavenet_vocoder/issues/7 ).
>
> - “The one result (the 4 numbers above) are conflated by both the change in model as well as utilizing the closed-form loss.”
> * The results in Table 1 (4 numbers) don’t utilize the closed-form loss. They are MOS results using different output distributions for autoregressive WaveNet. Our conclusion from Table 1 is that Gaussian WaveNet can produce competitive quality of samples as other options.
>
>  - “Writing-wise, the paper was clear, although potentially too packed with background information. As a expert on generative models, most of Sections 1-3 are already well-known and could be made more concise by referencing past works for more details. They add various details (such as the architecture notes at the end of 3.1) which should be better placed elsewhere to tease out what the important changes are in this paper."
> * Thanks for your nice suggestion. We have moved the architecture notes at the end of Section 3.1 to Appendix and experiment Section.  We have also shortened Section 3 in our revision.  Note that it is an application paper; a lot readers are from text-to-speech community. We referred past work for more details in Section 1-3, but we think a self-contained presentation with enough background information could be helpful to communicate with readers from different background.

---

### Official Review · AnonReviewer3 · 2018-11-09
**A solid and insightful experimental contribution to neural spectrum-to-waveform and speech synthesis (same rating after reviewing responses)**

**Rating:** 9
**Confidence:** 4

**Review:**

This paper proposes some modifications to established procedures for neural speech synthesis and investigates their effect experimentally. The proposed modifications are mostly fairly straightforward conceptually, but appear to work well, and this reviewer feels the paper has huge value in its experimental contributions extending and clarifying certain aspects of WaveNet training and distillation. The paper is well-written and fairly concise, with a short-and-sweet experimental results section.

Major comments:

The conceptual novelty seems a little overstated in the abstract. For example, the value seems to not really be in the "proposing" a text-to-wave neural architecture for speech synthesis (which, aside from important experimental tweaks, is essentially Tacotron 2 training all parameters from scratch) but in showing that it works well experimentally. Conceptually the paper is extremely close to the parallel wavenet paper, the main differences being slightly different component distributions (Gaussian instead of logistic), a different set of loss terms in addition to the reverse KL, and joint training of the spectral synthesis and waveform synthesis parts of the model.

It would be super insightful to include log probabilities on the test set (everywhere MOS results have been reported) in the experimental results. This would help tease apart the effects of architecture inductive bias, different divergences, distillation, etc. One of the really nice things about flow-based models is the ability to compute the log probability tractably.


Minor comments:

Perhaps mention that teacher forcing is maximum likelihood in the introduction? Currently it almost sounds like the paper is contrasting teacher forcing for WaveNet (paragraph 2) and MLE (list item 1).

At the end of paragraph 3 in the introduction, it would be helpful to mention that the intractable KL divergence being referred to is the frame-level one-step-ahead predictions, not the entire sequence-level prediction. Also, for 1D distributions isn't taking a large number of samples quite effective in practice?

In introduction list item 3, suggest mentioning Tacotron 2 (Shen et al) and contrasting with the present work for clarity.

In section 3.1, it surprises me slightly that clipping at -7 is essential. It would be helpful to state what exactly goes wrong if this is not done. Does it lead to overfitting and so bad test log likelihoods? What effect is noticeable in the generated samples?

Equation (6) is incorrect. It should be conditioned on < t, not <= t. Conditioning on z <= t would make x_t deterministic.

Equation (7) is technically true as written, but only because all the distributions involved are deterministic. If <= t is replaced with < t (which based on the mistake in (6) is what I suspect the authors intended) then it is no longer true. This equation is not used anywhere as far as I can tell. It seems to me like the property that enables non-recursive-over-time ("parallel") sampling is (5), not (7). Incidentally, when multiple one-step-ahead samples are taken per frame for parallel wavenet, the samples viewed at the sequence level are highly correlated, and do not obey anything like (7), but it doesn't affect the correctness of the expected value.

The IAF doesn't really "infers its output x at all time steps". Maybe "models" instead of "infers"?

Learning an "IAF directly through maximum likelihood" doesn't seem all that impractical. People train networks with recursive dependence such as RNNs (which is essentially what would be required to train certain forms of IAF with MLE) as opposed to non-recursive dependence such as CNNs all the time, after all. It seems like this claim depends on the details of the transform $f$.

Out of interest, did the authors consider reversing the sequence being generated in time between successive IAF blocks? This would limit the ability to do low latency synthesis but might improve performance considerably.

The first paragraph in section 3.3 seems like it should probably be part of section 3.3.1 (it's not related to other losses such as spectrogram frame loss, for example). It would be helpful to state explicitly that: (a) the goal is to minimize the sequence-level reverse KL; (b) this can be approximated by taking a single sample z, but this may have high variance; (c) the variance of this estimate can be reduced by marginalizing over the one-step-ahead predictions for each frame; (d) parallel wavenet's mixture of logistics means it has to use a separate Monte Carlo sampling at the frame-level, whereas the proposed Gaussian allows this one-step-ahead marginalization to be performed analytically. This one-step-ahead marginalization is an example of Rao-Blackwellization.

It didn't seem clear from section 3.3 and 3.3.1 that parallel wavenet also uses the one-step-ahead marginalization trick to reduce the variance.

It might be helpful to mention that using the reverse KL would be expected to have mode-fitting behavior, making samples sound better but log probability on the test set worse.

It was not clear to me what difference or similarity was being demonstrated in Figure 1.

Small point, but "Oord et al" should be "van Oord et al" throughout (it's a surname).

In section 3.3.2, can the authors give any insight as to why training with reverse KL alone leads to whispering, and why adding the STFT term fixes this? (If it's only something that's been noticed empirically, "will lead" -> "empirically we found"?)

I noticed quite a large qualitative perceptual difference between the student and teacher samples, particularly in the speech synthesis case (experiment 3), even though I think I'd rate the quality on a linear scale as fairly similar (in line with the MOS results). The teacher sounds noticeably "harsher" but "clearer" Do the authors have any insight as to why this perceptual difference occurs (if they also perceive a qualitative difference)? Is it probably a difference in inductive bias between an AF (which WaveNet can be seen as) and IAF?

I found it fascinating that reverse KL and forward KL lead to roughly the same MOS for spectrum-to-waveform. I assumed reverse KL would be better due to its preference for high-quality samples due to mode fitting.

Out of curiosity, what is responsible for the pops at the start of the spectrogram-conditioned distilled models? Also why are the synthesized samples shorter than the ground truth (less initial silence)?

---

> ### Author Response · Authors · 2018-11-26
> **To Reviewer 3 (part 1):**
>
>
> Thank you so much for the detailed comments and suggestions; they are really helpful to improve the quality of our paper.
>
> Major comments:
>
> - "The value seems to not really be in the "proposing" a text-to-wave neural architecture for speech synthesis (which, aside from important experimental tweaks, is essentially Tacotron 2 training all parameters from scratch) but in showing that it works well experimentally.”
> * Except training all parameters from scratch, our text-to-wave architecture is different from previous Tacotron 2 or Deep Voice 3, because the WaveNet vocoder is conditioned on the hidden states instead of mel-spectrogram from the encoder-decoder architecture. This difference is crucial to the success of training from scratch. Actually, we tried to simply connect text-to-spectrogram model and a mel-spectrogram conditioned WaveNet and train all parameters from scratch, but it performs worse than the separate training pipeline like Tacotron 2. We will emphasize this difference in our paper.
>
> - "It would be super insightful to include log probabilities on the test set (everywhere MOS results have been reported) in the experimental results."
> * Thanks for your nice suggestion. We will include the log probabilities results in our final draft. We also want share some preliminary observations here. We usually find that the test likelihood is not directly related to the quality of synthesized samples. For example, when we perform hyper-parameter search for autoregressive WaveNet, the validation likelihood is not reliable at all for selecting a “good” model that synthesizes high quality speech samples. This is probably the reason that test likelihood is not a common evaluation metric in speech synthesis community.
>
> Minor comments:
>
> - "Perhaps mention that teacher forcing is maximum likelihood in the introduction? Currently it almost sounds like the paper is contrasting teacher forcing for WaveNet (paragraph 2) and MLE (list item 1). "
> * Thanks for your suggestion. In contrast to the quantized surrogate loss for mixture of logistic distribution in Parallel WaveNet, we apply MLE for Gaussian. All autoregressive models are trained with teacher forcing. We have clarified it at list item 1 in our revision.
>
> - "At the end of paragraph 3 in the introduction, it would be helpful to mention that the intractable KL divergence being referred to is the frame-level one-step-ahead predictions, not the entire sequence-level prediction. Also, for 1D distributions isn't taking a large number of samples quite effective in practice? “
> * Thanks for your suggestion. In our draft, frame-level refers to STFT frame, so we add “intractable per-time-step KL divergence” at the end of paragraph 3.  In addition, we further clarify this point in Section 3.1 following your suggestion.  Monte Carlo sampling can be effective for 1D distribution, but it is certainly less effective than closed-form computation and may require a large number of samples for highly peaked distributions, which is usually the case for WaveNet. In practice, a large number of samples may also raise out-of-memory issue.
>
> - “In introduction list item 3, suggest mentioning Tacotron 2 (Shen et al) and contrasting with the present work for clarity.”
> * Thanks for your suggestion; we have mentioned Tacotron 2 and compared it with our work in list item 3.
>
> - "In section 3.1, it surprises me slightly that clipping at -7 is essential. It would be helpful to state what exactly goes wrong if this is not done.”
> * Yes, clipping is very important to avoid numerical problem (NaN) during training. When we track the NaN in our initial implementation, we found that $\sigma$ can be very small at some time-steps, which may lead to numerical issues.
>
> - "Equation (6) is incorrect. It should be conditioned on < t, not <= t. Conditioning on z <= t would make x_t deterministic. Equation (7) is technically true as written, but only because all the distributions involved are deterministic.
> * Many thanks for pointing it out. We have revised this notation error throughout the paper. To avoid confusion, we have also removed Eq. (7) and misleading description.
>
> - “The IAF doesn't really "infers its output x at all time steps". Maybe "models" instead of "infers"?”
> * Yes, we have changed it to “models” in revision.
>
> - "Learning an "IAF directly through maximum likelihood" doesn't seem all that impractical.”
> * Yes, we agree on that. WaveRNN is a good example. We have moderated our text.
>
> - "Out of interest, did the authors consider reversing the sequence being generated in time between successive IAF blocks?”
> * This is an excellent idea. We didn’t try it, but we will definitely try it afterwards.

---

> > ### Comment · AnonReviewer3 · 2018-12-05
> > **Response to authors' comments.**
> >
> > Thanks so much to the authors for thoroughly taking into account and properly integrating many of my suggestions. I personally feel like this is an even stronger paper now because of the changes.
> >
> > Very minor responses:
> >
> > I would indeed expect the log probabilities on the test set not to correlate very well with perceived sample quality, but that it would provide complementary information. Sample quality is measuring something like whether waveforms likely under the model are also likely in reality, whereas log probability is measuring something like whether waveforms likely in reality are also likely under the model. This is particularly relevant here since the original model is trained with KL divergence and the distilled model is trained with reverse KL, which optimize for different aspects of this trade-off. I would also expect log probability not to correlate with sample quality since a medium amount of statistical overfitting is very bad for log probability but often quite beneficial for sample quality.
> >
> > I think the "one-step-ahead" and "per-time-step" additions greatly clarify what is done and its differences to previous work. Thanks for including that.
> >
> > The first paragraph in section 3.3.1 now seems super clear to me.
> >
> > "Figure 1 implies a fast and persistent matching of $log \sigma$ in teacher and student models". That makes sense; thanks for the explanation. Perhaps consider also including the empirical histogram of the student log stdev when the unregularized reverse KL is used for distillation, to show the benefit of the proposed method.
> >
> > Obviously it's good to keep the paper concise, but I personally think the point about qualitative difference in sample quality between teacher and student, and the point about high-frequency blurriness, are interesting and worth mentioning in the paper, even if we don't yet understand exactly why it occurs. Up to the authors whether they think this is insightful or not, though.

---

> > > ### Comment · AnonReviewer3 · 2018-12-06
> > > **Minor fix**
> > >
> > > Just realized it should be "van den Oord" not "van Oord" by the way. Apologies for not catching that sooner.

---

> ### Author Response · Authors · 2018-11-26
> **To Reviewer 3 (part 2):**
>
>
> - “The first paragraph in section 3.3 seems like it should probably be part of section 3.3.1. It would be helpful to state explicitly that: (a) the goal is to minimize the sequence-level reverse KL; (b) this can be approximated by taking a single sample z, but this may have high variance; (c) the variance of this estimate can be reduced by marginalizing over the one-step-ahead predictions for each frame; (d) parallel wavenet's mixture of logistics means it has to use a separate Monte Carlo sampling at the frame-level, whereas the proposed Gaussian allows this one-step-ahead marginalization to be performed analytically.”
> * Many thanks for your great suggestion. We have reorganized Section 3.3 and stated (a)(b)(c)(d) explicitly in our revision.
>
> - “It was not clear to me what difference or similarity was being demonstrated in Figure 1.”
> * Figure 1 implies a fast and persistent matching of $log \sigma$ in teacher and student models because of the proposed regularization term, which is crucial to avoid numerical issue.  More importantly, monitoring the empirical histogram of $log \sigma$ during distillation is very helpful for reproducing ClariNet, because a successful distillation process always exhibits the empirical histograms like Figure 1.
>
> - “Small point, but "Oord et al" should be "van Oord et al" throughout (it's a surname).”
> * Thanks for your correction. We have fixed it throughout the paper.
>
> - “In section 3.3.2, can the authors give any insight as to why training with reverse KL alone leads to whispering, and why adding the STFT term fixes this? (If it's only something that's been noticed empirically, "will lead" -> "empirically we found"?)”
> * This is a very good question. We only have some intuitions behind this empirical observation, but we recommend a new ICLR submission which gives an in-depth analysis and provides non-trivial insights on this problem ( https://openreview.net/forum?id=rygFmh0cKm ). Adding STFT term fixes the whispering, because it will raise the energy of synthesized voice. We have changed “will lead” to “especially we found” in the revision.
>
> - “I noticed quite a large qualitative perceptual difference between the student and teacher samples, particularly in the speech synthesis case (experiment 3), even though I think I'd rate the quality on a linear scale as fairly similar (in line with the MOS results). The teacher sounds noticeably "harsher" but "clearer" Do the authors have any insight as to why this perceptual difference occurs (if they also perceive a qualitative difference)? Is it probably a difference in inductive bias between an AF (which WaveNet can be seen as) and IAF?”
> * Yes, we also perceive this qualitative perceptual difference.  When we visualize the spectrograms of student and teacher samples, we found that the high frequency bands of student samples tend to be more blurred than teacher’s.  It implies that the AF may be better at modeling the high frequency details than the non-autoregressive IAF.
>
> - "Out of curiosity, what is responsible for the pops at the start of the spectrogram-conditioned distilled models? Also, why are the synthesized samples shorter than the ground truth (less initial silence)? "
> * The synthesized samples are shorter than the ground truth, because our data preprocessing pipeline chopped the initial and trailing silence. It is also responsible for the pops at the start of the synthesized audios, because the model didn’t see enough silence at the start of audios during training. We will remove this problematic operation and update all synthesized samples afterwards.
>
> Many thanks again for your in-depth review and very insightful suggestion.

---

### Meta-Review · Area_Chair1 · 2018-12-14
**Well written paper with detailed experiments**

**Confidence:** 4
**Recommendation:** Accept (Poster)

**Metareview:**

The authors discuss an improved distillation scheme for parallel WaveNet using a Gaussian inverse autoregressive flow, which can be computed in closed-form, thus simplifying training. The work received favorable comments from the reviewers, along with a number of suggestions for improvement which have improved the draft considerably. The AC agrees with the reviewers that the work is a valuable contribution, particularly in the context of end-to-end neural text-to-speech systems.